# Level of Trust of the Population in the Unified Health System in the Midst of the COVID-19 Crisis in Brazil

**DOI:** 10.3390/ijerph192214999

**Published:** 2022-11-15

**Authors:** Juliana Soares Tenório de Araújo, Felipe Mendes Delpino, Thaís Zamboni Berra, Heriederson Sávio Dias Moura, Antônio Carlos Vieira Ramos, Murilo César do Nascimento, Jonas Bodini Alonso, Ruan Víctor dos Santos Silva, Giselle Lima de Freitas, Titilade Kehinde Ayandeyi Teibo, Roxana Isabel Cardozo Gonzales, Ricardo Alexandre Arcêncio

**Affiliations:** 1College of Nursing, University of São Paulo at Ribeirão Preto, Avenida dos Bandeirantes, 3900, Monte Alegre, Ribeirão Preto 14040-902, SP, Brazil; 2Nursing School, Federal University of Alfenas, Minas Gerais, Belo Horizonte 37048-395, MG, Brazil; 3Departament of Nursing, Federal University of Minas Gerais, Belo Horizonte 31270-901, MG, Brazil; 4Departament of Nursing, Federal University of Goiás, Goiânia 74690-900, GO, Brazil

**Keywords:** COVID-19, confidence, health services, basic health unit, hospital, public policy

## Abstract

Objective: to assess the level of trust in health services during the COVID-19 pandemic in Brazil. Methods: A cross-sectional study, carried out between 2020 and 2021, among Brazilians over 18. Nonprobabilistic sampling was used. Descriptive and inferential statistics were applied, using the local bivariate Moran’s technique to verify the existence of spatial dependence between the incidence and mortality of COVID-19 and trust in health services. Furthermore, multinomial regression was also used to analyze the factors associated with the confidence level, with the calculation of the odds ratio and with a confidence interval of 95%. Results: A total of 50.6% reported trust in hospital services, while 41.4% did not trust primary health care services. With the application of the local bivariate Moran, both for the incidence and mortality of COVID-19, the trust in tertiary care and primary care services showed a statistically significant spatial association predominant in the Midwest (high–low) and North (low–high) regions of Brazil. The level of trust was associated with education, religion, region of the country and income. Conclusions: The level of trust in hospital services, more than primary health care services, may be related to the population’s culture of prioritizing the search for hospital care at the detriment of health promotion and disease prevention.

## 1. Introduction

More than two years ago, the World Health Organization (WHO) declared COVID-19—a disease caused by SARS-CoV-2 virus—as a pandemic, which has since devastated and significantly burdened health systems, especially in low- and middle-income countries [1].

The pandemic has brought the most diverse problems to the country, directly influencing health services. The displacement of outpatient and elective care and the increase in waiting for diagnoses and treatments have generated dissatisfaction among the population that need to use such services. Dissatisfaction may be linked to trust in health services, which particularly during emergencies is essential for maintaining effective care [2].

Research in Brazil carried out in 2020 showed that 54% of the population studied trusted the health system of the country. Compared to other surveys, these were the lowest levels of confidence of the Brazilian population [3].

Understanding the level of trust in the era of COVID-19 is essential for a more positive effect, especially for collective actions to face the pandemic, such as adherence to the vaccine, use of masks, and social distancing.

Through searches of databases, considering descriptors of population, confidence level, COVID-19 and health services, no studies were found that addressed the subject in Brazil. Despite this, trust in health systems is an essential issue to produce evidence and contribute to the formulation of public policies and awareness for health managers. In this way, the present study aims to assess the population’s level of trust in health services during the COVID-19 pandemic in Brazil.

## 2. Materials and Methods

### 2.1. Design

This is a cross-sectional survey, carried out in a hybrid modality between August 2020 and February 2021. The study was conducted in accordance with the recommendations of the “Strengthening the Reporting of Observational Studies in Epidemiology” [4]. As for the sample design, the study was based on convenience sampling, using the snowball technique, which is characterized by the nonadoption of margin of error calculations, since it is a nonprobabilistic method [5].

The survey was publicized, and participants were invited to respond to the questionnaire through the websites of the participating research institutions, e-mail, WhatsApp^®®®^, social media (Facebook^®®®^, Instagram^®®®^, Twitter^®®®^), or blogs, in addition to invited people from the network and contacts of the researchers involved. In addition, the participants were instructed to recruit other people within their social circle to participate in the research, aiming to obtain the expected sample.

### 2.2. Research Scenario and Epidemiological Context

The pandemic made clear the health challenge to be faced. The insufficiency of adequate means to face the health crisis evidenced in the country’s information networks culminated in an exorbitant number of cases and deaths from COVID-19. The epidemiological data of COVID-19 in Brazil are alarming: 34.4 million confirmed cases, more than 683 thousand deaths, and an incidence rate of 16,383.7 per 100 thousand inhabitants, second only to the United States of America and India [1,2].

### 2.3. Population

As the study population, we considered the general Brazilian population: those who declared themselves resident in Brazil and foreigners (immigrants) who have resided in the country for at least 6 months. Both were 18 years of age or older.

As inclusion criteria, participants who responded to the dependent variable (trust in health services) were considered, and as exclusion criteria, participants who did not respond to the date of birth field were considered. The independent variables considered in the study are described in Table 1.

### 2.4. Data Analysis

Exploratory data analysis was performed using descriptive statistics, position calculation and measures of central tendency. For qualitative variables, absolute and relative frequencies were counted. Pearson’s chi-square test was used to analyze the association between the outcomes and the categorical variables of the study, while the Kruskal–Wallis test was used to compare the groups regarding numerical variables.

To perform the multinomial logistic regression analysis, the population’s trust in health services in the face of COVID-19 was considered as the outcome variable of the study. As the dependent variables of the study are ordinal, we chose to use the cumulative logic model.

The variables were selected in two stages. First, the presence of multicollinearity between the independent variables was evaluated, through the variance inflation factor measure, which is the most used. As the independent variables are the same for both outcomes, the result of this analysis will also be valid for both [6,7].

In the second stage, the variable selection procedure, called stepwise, was applied, using the Akaike information criterion (AIC) [8,9] for the other variables of the first stage. This procedure was applied considering the proportional probability model for numerical reasons related to the number of parameters. For the variables selected by the AIC, the assumption that the model parameters were not proportional was tested individually by the likelihood ratio test [8,10].

In the final model, the corresponding odds ratio (OR) was calculated from the parameters of the fitted models. The final nonconsecutive models were separated into: a little confident (1), confident (2), and very confident (3). All analyses were performed using the R program (https://www.r-project.org/about.html) with a significance level of 5% (α = 0.05) using the VGAM package [11]. Please note that the database and the script used are available as Appendix A.

To perform the spatial analysis, we chose to use all 5568 Brazilian municipalities as the unit of analysis, and the shapefile with the information was obtained through the form the Brazilian Institute of Geography and Statistics (IBGE) [12], for which the incidence and mortality rates of COVID-19 were calculated between the years 2020 and 2021. The number of cases was the numerator, while the population of the municipality was the denominator and the multiplication factor was per 1000 inhabitants. Regarding the calculations of rates, the number of cases and deaths by COVID-19 was obtained through the Coronavirus Panel and population data were extracted from the Brazilian Institute of Geography and Statistics (IBGE).

The bivariate Moran index [13] indicates the degree of association (positive or negative) between the value of the variable of interest in a given region and another variable in the same region, and in this way it is possible to map the statistically significant values, generating a map choropleth according to its classification. In the present study, we tested the spatial autocorrelation between the incidence and mortality rates of COVID-19 with the trust in primary (basic health unit) and tertiary (hospital) health services.

The classification can be given as: high–high (high values of incidence or mortality by COVID-19 with a high level of trust in health services (primary or tertiary)); low–low (low values of incidence or mortality by COVID-19 with a low level of trust in health services (primary or tertiary)); low–high (low values of incidence or mortality by COVID-19 with a high level of trust in health services (primary or tertiary)) and high–low (high values of incidence or mortality by COVID-19 with low level of trust in health services (primary or tertiary)). It is noteworthy that the high and low values were classified according to the average of the values of the variables of the neighboring regions [13].

## 3. Results

The study sample consisted of 1018 people, most of them female (75.2%; *n* = 766), with a mean age of 46.1 years. Most study participants were from the Southeast and South regions: 66% (*n* = 672) and 14.1% (*n* = 144), respectively. A total of 69.5% (*n* = 708) of the sample declared themselves to be white, 49.8% (*n* = 507) were married or in a stable union, and 37.8% (*n*: 385) were working as a public servant. A total of 54.6% (*n* = 556) had a graduate degree, 29.3% (*n*: 298) had a monthly income of 5 to 10 minimum wages, 71.3% (*n* = 726) were living in their own home, and 97.1% (*n* = 988) were living in urban areas. Approximately 87.6% (*n* = 892) of respondents did not receive any government assistance, while only 9.2% (*n* = 94) benefited from pandemic rescue assistance.

Of the total sample, about 77.3% (*n* = 787) had a health plan and used the SUS, while 85.1% (*n* = 866) did not receive a visit from the community health agent and 84.9% (*n* = 864) reported having a health center in their community or neighborhood (Table 2).

About 50.6% (*n* = 515) of the total sample said they trust hospitals and 41.4% (*n* = 421) said they had little trust in basic health units and health centers (Table 3).

Participants were asked about using sources of information to stay informed about COVID-19, and 62.8% responded that they consult materials from national or international organizations, 67.3% from materials produced by universities, 41% from social networks, and 60.1% stated that they also seek information via the internet. It is noteworthy that the questions regarding the sources of information had dichotomous answers (e.g., do you use the internet? Yes or no), and therefore, the percentages expressed refer to the number of people who claimed to use a certain source of information, but they are not exclusive options, that is, the person may have answered “yes” to more than one question.

Figure 1 shows the application of the local bivariate Moran index, which refers to trust in tertiary health services (Figure 1A,B); there is a statistically significant spatial association predominant in the Midwest regions (high–low), with high rates of incidence and mortality due to COVID-19 with low level of trust in health services. In the North region of the country (low–high) the opposite is noted, i.e., low rates of incidence and mortality by COVID-19 with high level of trust in health services.

The same pattern is observed in Figure 1C,D, referring to trust in primary health services, in which there is also a statistically significant spatial association predominant in the Midwest regions, with the same high–low pattern; that is, high rates of incidence and mortality due to COVID-19 with a low level of trust in health services. In the North region of the country, there is an opposite pattern, low–high, showing low rates of incidence and mortality due to COVID-19 with a high level of trust in health services.

Table 4 shows the final adjusted models, which corresponded to the level of trust in hospital institutions, including tertiary hospitals, field hospitals, and emergency care units (UPA) in the context of the COVID-19 pandemic. It was found that residents of the Midwest region were 50.02% (95%CI: 0.26; 0.95) less likely to respond as confident than residents of the Southeast region.

Residents of the North region were 56.87% (95%CI: 0.24; 0.74) less likely to respond as confident than residents of the Southeast region. On the other hand, residents of the South region were 2.07 (95%CI: 1.34; 3.17) times more likely to respond as confident and 3.18 (95%CI: 1.11; 9.11) times more likely to respond with little confidence compared to residents of the Southeast region.

For each year of life, there was a reduction of 2.48% (95%CI: 0.96; 0.98) in responding as less confident, confident, or very confident. People of evangelical religion were 34.64% (95%CI: 0.43; 0.98) less likely to respond as less confident, confident, or very confident than participants of the Catholic Christian religion. The student occupation was associated with 2.09 (95%CI: 1.13; 3.85) times more likelihood to respond as less confident, confident, or very confident.

Participants with elementary schooling or less were found to be 3.06 (95%CI: 1.29; 7.22) times more likely to respond as less confident, confident, or very confident than those with higher education. Regarding the location of the residence, participants from rural areas were 54.67% (95%CI: 0.22; 0.91) less likely to respond as less confident, confident, or very confident than participants who live in urban areas.

Participants who sought information from international organizations were 1.35 (95%CI: 1.06; 1.72) times more likely to respond with little confidence than those who did not consult these sources. Participants who earn 1 to 2 times the minimum wage had 64.9% (95%CI: 0.20; 0.61) less likelihood in responding as less confident, confident, or very confident than participants who receive more than 10 minimum wages.

Our final models corresponding to primary health care services (Table 5) showed that 41% of the Brazilian population has little confidence in primary health care services in coping with COVID-19.

Separated marital status was associated with a lower chance of responding as less confident, confident, or very confident in relation to individuals who are married or in a stable union (OR: 0.67; 95%CI: 0.47; 0.95).

Participants who have a postgraduate degree were 1.83 times more likely to respond as less confident, confident, or very confident than those who had a degree. In the same way, those with elementary education or less were 2.40 (95%CI: 1.09; 5.25) times more likely to respond as less confident, confident, or very confident than graduates. Those who live in rental/letting houses were 1.46 (95%CI: 1.10; 1.92) times more likely to respond as less confident, confident, or very confident than those who own a residence.

## 4. Discussion

This study aimed to assess the population’s level of trust in the health system in providing care and support in the COVID-19 era in Brazil. The data were collected at the first moment of the pandemic when isolation measures were more restrictive, a lot of misinformation was circulating, and health institutions were undergoing adjustments in services. The results point to greater confidence in the services of (tertiary) hospitals, field hospitals, and emergency care units in the care of COVID-19 cases. There was also lower trust in primary care services, probably related to the low resolution of these health institutions in the country.

The present study showed that 30.7% of the participants trusted some health measure instituted by government agencies. This differs from the trust observed in the Organization for Economic Co-operation and Development (OECD) countries, where 51% of people said they trust their national government and 71% reported trust in their healthcare system [14]. In this sense, the average trust in the health system of OECD countries also exceeds the trust of Brazilians in hospital services (50.6%) and in basic health units (58.6%).

Low education, low income, and living conditions are proxies for social vulnerability. The most vulnerable population has less access to health services and greater difficulties in care, and suffers more prejudice and mistreatment in services, affecting the levels of trust of this population [15].

The results of this study confirm this evidence that people with elementary education had less confidence than those with higher education, and those who received between one and two minimum wages were less confident than those who received more than ten minimum wages.

The results indicated that the younger population are 2.09 times more likely to respond with less confidence. This was also demonstrated in a study in the Netherlands; during the acute phase of the COVID-19 pandemic, the rigorous public health measures imposed in that country generated confidence in the institutions involved in their elaboration and implementation, especially among participants aged 65 and over. That is, the impact on trust in the government and in institutions was greater among older people and those at greater risk of developing the severe form of the disease. In this study, people with less education trust health institutions 3.06 times less. As low education is an indicator of poverty, these people are more socially vulnerable, with less access to health services, and consequently they tend to trust less.

Although there is no statistical significance in the present study regarding race/color, lack of trust in health services remains a problem in individuals who identify as black, as pointed out by Deloitte Insights [16] Racist experiences in health care drive users away from services; 80% of people who had a negative experience do not return for health care.

Rebuilding trust between this population and health services contributes to improving equity in health. In 2005, the WHO Commission on Social Determinants (CSD) cited racism as an intermediate factor in the production of health inequities, so the State developed some affirmative policies for greater inclusion and equity [17]. Although these aspects were not the objective of the present study, it is essential to highlight that discrimination and racism are factors associated with a lower level of trust, as evidenced in other studies.

In Brazil, there are significant socioeconomic discrepancies. About 25% of Brazilians are on the poverty line, almost half of the population (46.6%) has only a basic level of education, and more than 70% depend exclusively on the Unified Health System; therefore, inequality assumes great importance in health issues and in the trust of the system [17].

According to the report prepared by CONASS (National Council of Health Secretaries) in 2003, 41.8% of the Brazilian population says that the SUS is a system that cannot be trusted, and when this assessment is based on education and income, it is shown that there is an inverse relationship between the level of education and positive evaluation of the SUS. According to the report, 64% of illiterates approve and trust SUS, while among people with higher education, only 39% trust. It was found that in relation to income, the lower the income, the better the evaluation and confidence in the SUS [18].

Compared to our study, 64.9% of participants who receive between one and two minimum wages have little trust in the SUS, and with less education, they are 3.06 times more likely to have little trust. This is a significant change when compared to the 2003 report.

Another finding from the current survey is that those who rent are more likely to be less trusting than those who live in their own home. As demonstrated in other studies, social determinants influence a more conservative assessment; the position of greater economic power is associated with a more reliable assessment, and vice versa [15].

The study also showed a relevant level of confidence in relation to the Brazilian Unified Health System in the serious health crisis caused by COVID-19. The SUS historically suffers from a significant social stigma, especially from the middle class, which has always questioned the quality of its services, and believe that services in private entities are of better quality. During the pandemic, however, it seems that this population appropriated the SUS more like a right, and they are starting to see a SUS that works, and appropriating it, which is different from a reality verified in 2018 [19].

Concerning religion, evangelicals trust health institutions more, which follows the pattern of trust in the actions of the federal government, considering issues of political ideology. Although Brazil is a secular state, religious orders interfere in politics and ethical projects [20].

Information sources also influenced the trust of the population studied; people who consulted international organizations and official government institutions to stay informed about COVID-19 showed little trust in health services compared to those who did not. One possibility for such an outcome is that the wide availability of information through online media (official websites, medical websites, social networks, portals, blogs, forums, etc.), such as the spread of fake news, weakens the healthcare system worldwide [21]. The proper use of information technology tools plays an important role in strengthening confidence in information about COVID-19.

A key factor in facing the spread of the new coronavirus is mutual trust between people and their governments [22]. In this context, it is extremely important that citizens trust the recommendations they receive from public authorities.

This study revealed a varied pattern of trust of the Brazilian population in health institutions for the primary and tertiary care levels, which can be explained geographically, culturally, and according to religion and economic and social level. It has been shown that the population trusts hospital services more than primary care services, which may be related to emergency needs of health care during the first year of the pandemic.

We therefore emphasize the need to develop interventions in primary health care (PHC) to better solve health problems and consequently increase the population’s confidence in these services.

This study advances knowledge by highlighting the level of trust in health services during the first year of the COVID-19 pandemic. Still, as a reflection of a curative and hospital-centered model, the majority of the population has a culture of immediate care, to the detriment of the conditions of prevention and health promotion actions.

In Brazil, a health policy was adopted, prioritizing the structuring of the hospital network to care for the most serious cases, with little emphasis on disease prevention actions and the strengthening of primary care.

Popular education, a contact surveillance plan, and mass testing of the population, associated with the understanding that PHC is a low-cost ally in fighting and controlling an infectious disease, strengthen the health system.

We suggest the continuity of this in future studies, with greater diversity of the population sample, as well as the development of a longitudinal methodological design, since it is important for monitoring the evolution of the level of confidence in the various phases of the pandemic.

As limitations of the study, it is worth noting that the population was obtained using the snowball sampling technique, and due to this fact, the sample included in the study may differ from the profile of the Brazilian population. As it is a survey carried out during the pandemic (2020 and 2021) in online format, some social segments were not covered, such as people in situations of social vulnerability, which represents another limitation for this study. This justifies the reason that the research participants are mainly composed of people with a high level of education and higher income, as seen in other studies, in addition to the fact that people with higher education are the most participatory [23,24]. The involuntary exclusion of these groups may have influenced the estimates of the proportion of responses.

Another limitation refers to the impossibility of obtaining data on some aspects of the Brazilian population, such as socioeconomic class. Variables related to educational level and income, analyzed in isolation, may not be sufficient to characterize the profile of the population. Furthermore, secondary data related to the incidence and mortality rates by COVID-19 should be analyzed with caution, since the underreporting of the disease is a reality in Brazil, as well as the interpretations about the highs and lows. 

In this way, it is emphasized and encouraged to carry out further studies to understand the level of trust of the Brazilian population in health services during the COVID-19 pandemic in Brazil.

## 5. Conclusions

This study showed the level of trust of the Brazilian population in health institutions, verifying a varied pattern of trust for the levels of primary and tertiary care. The population trusts hospital services more than primary care services, which may be related to emergency care needs during the first year of the COVID-19 pandemic in the country.

Through spatial analyses, an association is demonstrated between trust in health services and the incidence of cases and the death rate. In regions where the population’s level of trust in health services was higher, there was a lower incidence of cases and deaths.

Thus, the development of interventions in primary care to better solve public health problems and consequently increase the population’s trust in these services has an impact on containing the spread of the virus.

Promotion and prevention measures must promote the engagement of society, so that citizens feel like partners in action and in the development of health policies, strengthening trust between authority and citizens.

## Figures and Tables

**Figure 1 ijerph-19-14999-f001:**
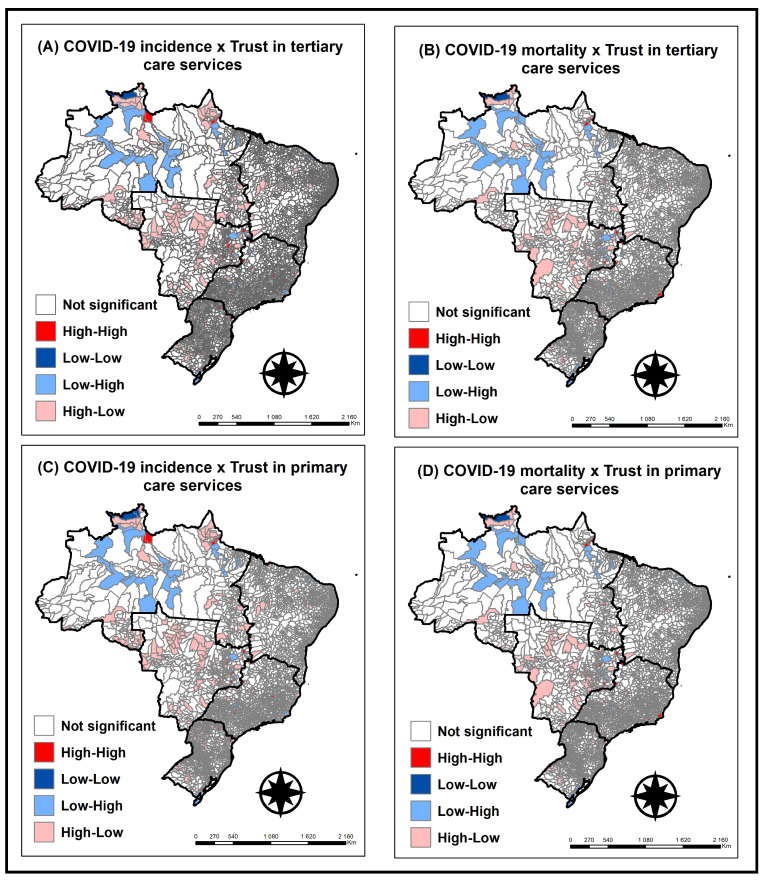
(**A**) Bivariate spatial dependence analysis (Moran index) between the COVID-19 incidence rate and the population’s level of trust in tertiary care health services. (**B**) Bivariate spatial dependence analysis (Moran index) between COVID-19 mortality rate and the population’s level of trust in tertiary care health services. (**C**) Bivariate spatial dependence analysis (Moran index) between COVID-19 incidence rate and the population’s level of trust in primary care health services. (**D**) Bivariate spatial dependence analysis (Moran index) between COVID-19 mortality rate and the population’s level of trust in primary care health services.

**Table 1 ijerph-19-14999-t001:** Variables included in the present study.

Variables	Categories
Independent Variables	
Sex	Male, Female
Age	Continuous
Region	North, Northeast, Midwest, Southeast, South
Education level	<=Elementary school, complete primary education, high school, graduate, and postgraduate degree
Income	Less than 1 minimum wage, from 1 to 2 minimum wages, from 2 to 5 minimum wages, from 5 to 10 minimum wages, above 10 minimum wages, don’t know/prefer not to inform
Skin color	White, black/brown, yellow/indigenous
Marital status	Single, married/stable union, separated, widowed
Religion	Catholic, spiritism, evangelical, other religions, no religion
Occupation	Retired, unemployed, public employee, private employee, self-employed/informal/offers, student, other
Years of residence	0 to 10 years, >10 to 20 years, >20 to 30 years, > 30 years
Number of people living at home	Continuous
Number of people contributing to the family income	Continuous
Area of residence	Urban, rural
Type of housing	Own, rented/leased, other
Health insurance	Yes, No
Use of SUS	Yes, No
Receives a visit from the community agent	Yes, No
Health clinic where he lives	Yes, no, I do not know
Aid from the government	Yes, No
Sources of information	
International organizations	Yes, No
Friends and/or family	Yes, No
Social support networks	Yes, No
News, newspapers	Yes, No

**Table 2 ijerph-19-14999-t002:** Sociodemographic characteristics of the final study sample (*n* = 1018).

Numerical Variables	Mean	Standard Deviation
Age	46.1	14.8
Residents at home	2.8	1.7
People who contribute to the family income	1.8	0.8
Categorical variables	Frequency	%
Sex		
Male	252	24.8
Female	766	75.2
Region		
Midwest	43	4.2
Northeast	94	9.2
North	65	6.4
Southeast	672	66
South	144	14.1
Color/race		
Yellow/indigenous	21	2.1
White	708	69.5
Black/brown	289	28.4
Marital status		
Married/stable union	507	49.8
Separated	143	14
Single	351	34.5
Windowed	17	1.7
Religion		
Catholic	389	38.2
Spiritism	122	12
Evangelical	114	11.2
Other	93	9.1
No religion	300	29.5
Occupation		
Retired	141	13.9
Autonomous/informal/offers	97	9.5
Unemployed	46	4.5
Private employee	146	14.3
Public employee	385	37.8
Student	124	12.2
Other	79	7.8
Education		
Incomplete elementary school	9	0.9
Complete primary education	16	1.6
High school	177	17.4
Graduated	260	25.5
Postgraduate studies	556	54.6
Income (in minimum wages)		
Less than 1	22	2.2
1 to 2	72	7.1
2 to 5	273	26.8
5 to 10	298	29.3
More than 10	286	28.1
Prefer not to inform/do not know	61	6
No income	6	0.6
Type of housing		
Own	726	71.3
Rent/Lease	242	23.8
Granted	37	3.6
Other	13	1.3
Area of residence		
Rural	30	2.9
Urban	988	97.1

**Table 3 ijerph-19-14999-t003:** Characteristics of the final study sample: social thermometer (*n* = 1018).

Categorical Variables	Frequency	%
Trust level in public health services to face COVID-19		
Hospital		
No trust	74	7.3
Little trust	326	32
Trust	515	50.6
High trust	103	10.1
Basic health unit/health center		
No trust	143	14
Little trust	421	41.4
Trust	396	38.9
High trust	58	5.7

**Table 4 ijerph-19-14999-t004:** Final models regarding the trust level toward hospital services (*n* = 1018).

	Odds Ratio (OR)	95%CI	*p*-Value
Sex			
Female	Ref	Ref	Ref
Male	1.36	1.02–1.81	0.0340
Region			
Southeast	Ref	Ref	Ref
Midwest: less confident	0.92	0.27–3.17	0.9032
Midwest: confident	0.49	0.26–0.95	0.0357
Midwest: very confident	0.15	0.02–1.15	0.0689
North East: less confident	2.18	0.75–6.33	0.1483
North East: confident	0.95	0.60–1.52	0.8589
North East: very confident	1,21	0.62–2.38	0.5665
North: less confident	0.52	0.22–1.20	0.1296
North: confident	0.43	0.24–0.74	0.0027
South: less confident	3.18	1.11–9.11	0.0309
South: confident	2.06	1.13–3.17	0.0010
South: very confident	0.90	0.49–1.65	0.7460
Age	0.97	0.96–0.98	0.0000
Religion			
Catholic	Ref	Ref	Ref
Spiritist	1.38	0.92–2.07	0.1162
Evangelical	0.65	0.43–0.98	0.0429
Other	0.68	0.44–1.07	0.1022
No religion	0.91	0.67–1.23	0.5562
Occupation			
Other	Ref	Ref	Ref
Retired	1.10	0.62–1.94	0.7363
Autonomous	1.24	0.70–2.20	0.4580
Private employee	0.93	0.46–1.91	0.8616
Public employee	0.84	0.49–1.45	0.5519
Student	1.48	0.91–2.39	0.1085
Residential area			
Urban	Ref	Ref	Ref
Rural	0.45	0.22–0.91	0.0267
Sources of information to stay informed about COVID-19			
International organization: No	Ref	Ref	Ref
International organization: Yes	1.41	1.09–1.82	0.0074
Education			
Graduated	Ref	Ref	Ref
Elementary school	3.05	1.29–7.22	0.0109
High school	1.15	0.77–1.71	0.4853
Postgraduate studies	1.32	0.98–1.78	0.0673
Income (in minimum wages)			
More than 10	Ref	Ref	Ref
1 to 2	0.35	0.20–0.61	0.0003
2 to 5	0.69	0.49–0.98	0.0436
5 to 10	0.64	0.47–0.89	0.0089
Less than 1	0.36	0.15–0.83	0.0178
Prefer not to inform/do not know	0.65	0.37–1.12	0.1247

**Table 5 ijerph-19-14999-t005:** Final models regarding the trust level toward primary care services (*n* = 1018).

	Odds Ratio (OR)	95%CI	*p*-Value
Marital status			
Married	Ref	Ref	Ref
Separated	0.67	0.47–0.95	0.0274
Single	1.17	0.89–1.52	0.2417
Windowed	1.55	0.61–3.90	0.3503
Education			
Graduated	Ref	Ref	Ref
Elementary school	2.40	1.09–5.25	0.0283
High school	1.06	0.73–1.52	0.7484
Postgraduate studies	1.83	1.38–2.42	0.0000
Type of housing			
Own	Ref	Ref	Ref
Rent/lease	1.45	1.10–1.92	0.0073
Other	1.51	0.88–2.60	0.1339
Sources of information used to stay informed about COVID-19			
International organizations: No	Ref	Ref	Ref
International organizations: Yes: 1	1.35	1.06–1.72	0.0151
Friends and/or family: No	Ref	Ref	Ref
Friends and/or family: Yes: 3	1.32	1.02–1.70	0.0312

## Data Availability

The data from this study are stored in the REDCAp Platform, in the domain of the University of São Paulo, and are applied to the availability of data, which were granted under license for the present study. The REDCap platform was chosen due to its security in terms of data storage; however, the data used in the study can be made available upon request of the authors and plausible justification.

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
