# Peer review of "Level of Trust of the Population in the Unified Health System in the Midst of the COVID-19 Crisis in Brazil"

_ijerph, 2022, doi:10.3390/ijerph192214999_

Round 1

Reviewer 1 Report

The article entitled “Level of trust of the population in the Unified Health System in the midst of the COVID-19 crisis in Brazil” by Araújo et al points out the levels of trust of population of Brazil citizens over health systems during Covid pandemic (2020 – 2021 survey). The manuscript design is well set, however, the section in results is very confusing (for an individual with dyslexic-like such as myself, e.g., Line 192 - 194). The authors are recommended to rephrase the sentences wherever applicable to remove the confusion in the flow of the manuscript. Further, I feel that the manuscript is suitable for publication after the following issues are addressed:

Minor

  1. More detailed description of how the data can be accessed via REDCAp should be mentioned in the ‘Data Availability Statement’.
  2. All the scripts used in this analysis should be made available on GitHub profile with datasets as R objects or as supplementary material.  
  3. Line 136, and throughout the manuscript the phrase ‘do not trust’ should be replaced with ‘Little trust’ as described in Table 3. The ‘do not trust’ literally means No or ‘Nothing trust’ which as I understand is incorrect comparison.
  4. Line 138 – 140., the percent doesn’t add up to 100%. Explain in more detail.
  5. From Table 4: What are the 1, 2 and 3 refer to in ‘Region’? e.g. South:1. The authors need to describe these (also, please refer to Table 1).
  6. Missing references in Table 4 and 5, must be addressed.

Author Response

Point 1: The article entitled “Level of trust of the population in the Unified Health System in the midst of the COVID-19 crisis in Brazil” by Araújo et al points out the levels of trust of population of Brazil citizens over health systems during Covid pandemic (2020 – 2021 survey). The manuscript design is well set, however, the section in results is very confusing (for an individual with dyslexic-like such as myself, e.g., Line 192 - 194). The authors are recommended to rephrase the sentences wherever applicable to remove the confusion in the flow of the manuscript.

Response 1: Thank you for evaluating our manuscript. As recommended by this reviewer, there has been rephrasing of the sentences from lines 192 - 194, as well as suitability throughout the manuscript. 

Point 2: More detailed description of how the data can be accessed via REDCAp should be mentioned in the ‘Data Availability Statement’.

Response 2: Thank you for your comment, more details have been inserted in the appropriate section

Point 3: All the scripts used in this analysis should be made available on GitHub profile with datasets as R objects or as supplementary material.  

Response 3: The script was inserted as supplementary material.

Point 4: Line 136, and throughout the manuscript the phrase ‘do not trust’ should be replaced with ‘Little trust’ as described in Table 3. The ‘do not trust’ literally means No or ‘Nothing trust’ which as I understand is incorrect comparison.

Response 4: Thank you for your comment. The manuscript has been corrected

Point 5: : Line 138 – 140., the percent doesn’t add up to 100%. Explain in more detail.

Response 5: It is noteworthy that the questions regarding the sources of information had dichotomous answers (eg: do you use the internet? Yes or no) and, therefore, the percentages expressed refer to the number of people who claimed to use a certain source of information, but they are not options exclusionary, that is, the person may have answered "yes" to more than one question. This was explained in the text

Point 6: From Table 4: What are the 1, 2 and 3 refer to in ‘Region’? e.g. South:1. The authors need to describe these (also, please refer to Table 1).

Response 6: Thank you for your comment. The manuscript has been corrected

Point 7: Missing references in Table 4 and 5, must be addressed.

Response 7: Thank you for your comment. The manuscript has been corrected

Reviewer 2 Report

Although the study objectives and conclusions are interesting and worth reading I wonder about the representativiness of the sample, particularly on gender distribution. Regarding expressed level of trust,  although it's clear that Hospital have a superior level of trust, concluding that "little trust" equals "no trust" (line 135) seams excessive.  

Author Response

Point 1: Although the study objectives and conclusions are interesting and worth reading I wonder about the representativiness of the sample, particularly on gender distribution. Regarding expressed level of trust,  although it's clear that Hospital have a superior level of trust, concluding that "little trust" equals "no trust" (line 135) seams excessive.  

Response 1: Thank you for your comment. In the methods section, more details about how the survey was advertised and participants were invited were included. In the discussion, paragraphs have been inserted about the limitations of the study, mainly regarding sampling.

Reviewer 3 Report

The article by de Araujo et al is interesting and timely.

In the methods, the authors describe support from the survey data. However, in any large country like Brazil, the surveys may not truly represent the population under consideration. Hence, I suggest considering the methods of limitations of large population surveys, for example [1] below.

Also, globally the underreporting of COVID-19 is enormous, at least from the modeling perspective. Hence, whatever the impact of COVID-19 that we discuss aftermath it only indicates very little on the ground that is been reported, see for example [2] below.

I hope the authors discuss these while revising their article.References:

1.     Rzewuska M, de Azevedo-Marques JM, Coxon D, Zanetti ML, Zanetti ACG, Franco LJ, et al. (2017) Epidemiology of multimorbidity within the Brazilian adult general population: Evidence from the 2013 National Health Survey (PNS 2013). PLoS ONE 12(2): e0171813. https://doi.org/10.1371/journal.pone.0171813

2.     SG Krantz, DA Swanson, ASRS Rao (2022). Global underreporting of COVID-19 cases during 1 January 2020 to 6 May 2022. Current Science 123 (6), 741-742

Author Response

Point 1: In the methods, the authors describe support from the survey data. However, in any large country like Brazil, the surveys may not truly represent the population under consideration. Hence, I suggest considering the methods of limitations of large population surveys, for example [1] below. Also, globally the underreporting of COVID-19 is enormous, at least from the modeling perspective. Hence, whatever the impact of COVID-19 that we discuss aftermath it only indicates very little on the ground that is been reported, see for example [2] below.

Response 1: Thank you for your comment. In the discussion, paragraphs have been inserted about the limitations of the study.